# Inverse Relationship between Mean Corpuscular Volume and T-Score in Chronic Dialysis Patients

**DOI:** 10.3390/medicina58040497

**Published:** 2022-03-30

**Authors:** Ming-Hsiu Chiang, Chih-Yu Yang, Yi-Jie Kuo, Chung-Yi Cheng, Shu-Wei Huang, Yu-Pin Chen

**Affiliations:** 1Department of General Medicine, Kaohsiung Chang Gung Memorial Hospital, Kaohsiung 833, Taiwan; b101103050@tmu.edu.tw; 2Division of Nephrology, Department of Medicine, Taipei Veterans General Hospital, Taipei 112, Taiwan; cyyang3@vghtpe.gov.tw; 3Institute of Clinical Medicine, School of Medicine, National Yang Ming Chiao Tung University, Taipei 112, Taiwan; 4Stem Cell Research Center, National Yang Ming Chiao Tung University, Taipei 112, Taiwan; 5Department of Orthopedics, Wan Fang Hospital, Taipei Medical University, Taipei 116, Taiwan; benkuo5@gmail.com (Y.-J.K.); judyya1022@gmail.com (S.-W.H.); 6Department of Orthopedics, School of Medicine, College of Medicine, Taipei Medical University, Taipei 110, Taiwan; 7Division of Nephrology, Department of Internal Medicine, Wan Fang Hospital, Taipei Medical University, Taipei 116, Taiwan; m105095015@tmu.edu.tw; 8Department of Internal Medicine, School of Medicine, College of Medicine, Taipei Medical University, Taipei 110, Taiwan

**Keywords:** kidney failure, chronic, renal dialysis, peritoneal dialysis, mean corpuscular volume, bone density

## Abstract

*Background and Objectives:* Osteoporosis and anemia are prevalent among chronic kidney disease stage 5D (CKD stage 5D) patients. Osteoblasts are known as the niche cells of hematopoietic stem cells (HSCs) and stimulate HSCs to form blood-cell lineages within bone marrow microenvironments. We hypothesized that an inverse correlation may exist between mean corpuscular volume (MCV), a surrogate for ineffective hematopoiesis, and bone mineral density (BMD) in the CKD stage 5D population. *Materials and Methods:* This is a cross-sectional designed cohort study evaluating CKD stage 5D patients who have received dialysis therapy for over three months. Baseline clinical characteristics and laboratory data were prospectively collected. The dual-energy X-ray absorptiometry (DXA) method was used to measure BMD at five sites, which were bilateral femoral neck, total hip, and lumbar spine 1–4. The Pearson correlation test was initially adopted, and a multivariate linear regression model was further applied for potential confounder adjustments. *Results:* From September 2020 to January 2021, a total of 123 CKD stage 5D patients were enrolled. The Pearson correlation test revealed a significant inverse association between MCV and BMD at bilateral femoral neck and lumbar spine. The lowest T-score of the five body sites was determined as the recorded T-score. After adjustments for several potential confounding factors, the multivariate linear regression model found consistent negative associations between T-score and MCV. *Conclusions:* The present study found significant inverse correlations between MCV and BMD at specific body locations in patients on dialysis. A decreased T-score was also found to be associated with macrocytosis after adjustments for confounding variables. However, direct evidence for the causative etiology was lacking.

## 1. Introduction

Osteoporosis and anemia are commonly found in patients on dialysis. These two comorbidities have long been considered to result from distinct mechanisms. Chronic inflammation status, parathyroid hormone abnormalities, and elevating circulating levels of fibroblast growth factor 23 (FGF23) could affect normal osteoblast function in chronic kidney disease stage 5D (CKD stage 5D) patients, resulting in osteoporosis [1]. On the other hand, deregulated iron metabolism is believed to impede erythroblast differentiation, which in turn leads to anemia in patients with chronic kidney disease (CKD) [2]. However, a link may exist between the pathogenesis of osteoporosis and anemia in patients on dialysis.

Hematopoietic stem cells (HSCs) are located within bone marrow microenvironments or niches and allow for the development of multiple blood-cell lineages [3]. In vivo, osteoblasts play a crucial role in hematopoiesis by comprising part of these niches and producing hematopoietic growth factors [4]. We hypothesized that the heavy inflammatory burden of CKD stage 5D causes imbalance in bone remodeling and derangement in cell signaling, which may not only interfere with osteoblast function but also result in the poor expansion of HSCs [5].

In the present study, we used mean corpuscular volume (MCV) as a surrogate of injury to the bone marrow microenvironment or ineffective hematopoiesis. MCV measures the mean volume of red blood cells (RBCs), and has long been an indicator of clinical conditions such as poor nutrition, iron and vitamin B12 deficiency [6], and drug abuse. A previous cohort study reported persistent MCV elevation after the performance of autologous HSCs transplantation [7]. An epidemiology study suggested that myelodysplastic syndromes and suspected bone marrow failure are the leading causes of macrocytic anemia [8]. Dual-energy X-ray absorptiometry (DXA) was performed to measure bone mineral density (BMD) and thus determine the correlation between MCV and BMD in a Taiwanese population with CKD stage 5D.

## 2. Materials and Methods

### 2.1. Study Design

From September 2020 to January 2021, patients who were diagnosed as having CKD stage 5D and were on dialysis at a single medical center in Taipei, Taiwan, were prospectively enrolled in this cross-sectional study. Patients were included if they were aged ≥18 years, diagnosed as having CKD stage 5D (regardless of underlying etiologies or dialysis methods), and had been on maintenance dialysis for at least 3 months. Patients were excluded if they underwent kidney transplantation, experienced active bleeding (such as gastrointestinal bleeding resulting in hospitalization), or received a blood transfusion in the week before their scheduled DXA examination. The study was conducted in accordance with the code of ethics of the World Medical Association (Declaration of Helsinki) and was approved by the Ethics Committee of Taipei Medical University (N202103059). All participants provided written consent to their participation and the publication of their data.

All the eligible patients underwent DXA assessment for BMD and body composition indicators such as total fat percentage and skeletal muscle mass. Basic demographic data, including age, sex, body-mass index (BMI), and underlying comorbidities, were collected for analysis. Laboratory data obtained within the 2 months prior to the participants’ DXA examination were also collected; these data included the participants’ hemoglobin level (Hb), RBC count, red cell distribution width (RDW), platelet count and white blood cell (WBC) count, and levels of iron, ferritin, sodium, potassium, calcium, phosphate, parathyroid hormone, serum albumin, cholesterol, triglyceride, and liver enzymes. Furthermore, the participants’ smoking and dialysis statuses (in the 2 months prior to their DXA examination) were retrieved from their clinical records.

### 2.2. DXA Assessment 

All the participants underwent a DXA assessment on bilateral total hip, femoral neck, and lumbar spine (Prodigy, GEHC Lunar, Madison, WI, USA) for BMD. Scanning was performed by certified radiographers who were blinded to the trial design, and the imaging machine was calibrated daily as per manufacturer recommendations.

### 2.3. BMD

BMD (expressed in gm/cm^2^) was measured at the lumbar spine 1–4, bilateral femoral neck, and total hip. The recorded T-score was the lowest T-score of these five regions; this allows for a BMD comparison with healthy young (aged 20) adults [9].

### 2.4. Diagnosis of Osteoporosis

The World Health Organization criteria for diagnosing osteoporosis were adopted. Osteopenia corresponded to a T-score of more than −2.5 standard deviation (SD) but less than −1 SD. Osteoporosis was defined as a T-score less than or equal to −2.5 SD for at least one of the following sites: lumbar spine, femoral neck, and total hip [10].

### 2.5. Statistical Analysis

All statistical analyses were conducted using SPSS for Windows version 18.0 (SPSS, Chicago, IL, USA). Continuous variables are presented as means ± SDs, whereas categorical variables are presented as frequencies and numbers (percentage). Pearson correlation tests were conducted to analyze the relationship between MCV and baseline clinical characteristics or DXA results. When BMD was verified through the Pearson correlation test to be significantly correlated with MCV, this correlation was further analyzed through multivariate linear regression. Three models were employed with MCV or T-score as the dependent variable; these models consisted of several clinical confounders that were either highly related to T-score or MCV detected by univariate Pearson correlation tests. Model 1 was adjusted for clinical variables with significant (*p* ≤ 0.05) or marginally significant (*p* ≤ 0.1) correlations with MCV and Model 2 was adjusted for covariates with significant (*p* ≤ 0.05) or marginally significant (*p* ≤ 0.1) correlations with T-score. Model 3 was adjusted for several confounding variables reported in the previous literature that were highly correlated with both MCV and T-score. For the subgroup analysis of hemodialysis (HD) and peritoneal dialysis (PD) patients, independent t-tests and chi-squared tests were performed to compare continuous and categorical variables, respectively. For all tests, a two-sided *p*-value of <0.05 was considered statistically significant. 

## 3. Results

In total, 123 patients (61 men [49.6%] and 62 women [50.4%]) who had CKD stage 5D and were undergoing regular dialysis were enrolled. Their mean age and dialysis duration were 65 ± 11 years (age range of 40.3–90.3 years) and 59 ± 37 months, respectively. Ninety patients (73%) received HD three times per week, and the other 33 participants (27%) received continuous PD. The most prevalent comorbidities were hypertension (80%), diabetes mellitus (51%), and dyslipidemia (33%). Approximately 20% of the enrolled patients had a smoking history or were current smokers (Table 1).

Table 2 presents the DXA results; the mean T-score was −2.5 ± 1.2, indicating that over half of the patients were diagnosed as having osteoporosis (55%); significant inverse correlations with MCV were found in T-score (*p* = 0.003, β = −0.27), BMD at bilateral femoral head (right side: *p* = 0.008, β = −0.24; left side: *p* = 0.02, β = −0.21), and BMD at lumbar spine 1–4 (*p* = 0.038, β = −0.19). After calculations were performed using FRAX TAIWAN, the average 10-year probability rates of major osteoporotic fracture and hip fracture were revealed to be 11 ± 9.1% and 5.4 ± 7.4%, respectively.

Most patients’ MCV values were in the normal range, which is 85 to 100 fL (80.5%); 14 and 10 patients had MCV values that were less than 85 fL and more than 100 fL, respectively (Figure 1). MCV was revealed to be highly correlated with age, Hb, RBC count, RDW, serum phosphate, ferritin, and cholesterol levels. In terms of T-score, age, BMI, serum phosphate, parathyroid hormone level, gastrointestinal ulcer history, and dyslipidemia were identified as significant confounding variables (Appendix A). The correlation between T-score and MCV was investigated using three multivariate linear regression models. Clinical variables that were significantly associated with T-score or MCV in Appendix A were served as covariates in the multivariate linear regression model. In Model 1, MCV was set as the dependent variable and clinical factors that were highly correlated with MCV, including T-score, age, serum phosphate level, RDW, Hb, WBC, iron, ferritin, and cholesterol, were the covariates. Significant inverse correlation between MCV and T-score was observed after adjustments were made (*p* = 0.037, β = −0.96; Table 3). In Model 2, T-score was set as the dependent variable and adjustments were made for MCV, age, serum phosphate level, BMI, parathyroid hormone level, dyslipidemia, and gastrointestinal tract ulcer history. T-score was independently correlated with MCV (*p* = 0.048, β = −0.027; Table 3). Previous literature has indicated age, chronic inflammation, nutrition status, and liver dysfunction significantly influences both MCV and BMD [11,12]. Therefore, we performed a third multivariate linear regression model, in which T-score was set as the dependent variable and adjusted by MCV, age, WBC count, albumin, and liver enzymes (glutamate oxaloacetate transaminase, GOT; glutamate pyruvic transaminase, GPT). Results of Model 3 revealed significant inverse correlation between T-score and MCV (*p* = 0.036, β = −0.03; Table 3).

We performed a subgroup analysis to determine whether dialysis type influenced the results. For baseline clinical characteristics, the HD population had a higher levels of serum albumin relative to the PD population. By contrast, patients on PD had higher corrected calcium values, and iron levels relative to patients on HD (Table 1). Pearson’s correlation test revealed that the HD cohort had significant inverse correlations between T-score and MCV in each model (*p* values of 0.004, 0.04, and 0.036 for Models 1, 2, and 3, respectively; β values of −0.97, −0.036, −0.04 for Models 1, 2, and 3, respectively); by contrast, no significant associations were observed between MCV and T-score in patients on PD (Appendix A).

## 4. Discussion

The present study examined the clinical data of patients who had stable CKD stage 5D and were on chronic dialysis, and a higher MCV was revealed to be associated with lower T-score. This inverse correlation was further corroborated after adjustment by different confounding parameters in three different models.

Macrocytosis has been investigated extensively in hematology, and it has attracted increasing attention from other specialties in the last decade. Macrocytosis is related to higher mortality and morbidity in patients with cardiovascular diseases [13,14]. It is also a significant predictor of cancer relapse or all-cause mortality in patients with several types of gastrointestinal cancers [15,16,17]. For the CKD stage 5D population, Tennankore et al. reported that having an MCV of >102 fL is associated with higher all-cause mortality within 9 months of follow-up [18]. A similar finding was also reported by Dratch et al., who conducted a retrospective observational cohort study to investigate the clinical data of patients on incident HD [19]. The underlying attributable mechanism is unclear; however, several etiologies, including systematic inflammation, malnutrition, and altered hematopoiesis, have been proposed. The exact mechanism underlying the significant inverse association between MCV and T-score has not been clarified, and we hypothesize that the HSC niche may play a crucial role in this mechanism. In 1994, Taichman et al. proposed that osteoblasts participate in hematopoiesis through the production of granulocyte colony-stimulating factors [20]. Subsequently, Zhang et al. and Calvi et al. conducted animal research that verified the role of osteoblasts in regulating HSC formation; Zhang et al. examined bone morphogenetic protein signals, whereas Calvi et al. investigated the Notch signaling pathway. Both teams applied genetics-based strategies to demonstrate concomitant increases in HSCs and osteoblast populations in genetic mutant mice and to verify that the targeted deletion of osteoblasts results in a subsequent loss of HSCs [4,21]. Osteoblast activity was disturbed in patients with CKD stage 5D and therefore, CKD may directly affect the integrity of the HSCs niche. An in vitro study corroborated this hypothesis. Aleksinskaya used mice that underwent nephrectomy to emulate CKD and discovered that their HSCs population were reduced, and bone marrow microenvironments were functionally impaired compared with those of the control group [22].

It is interesting to note that BMD at the lumbar spine and bilateral femoral neck had significant inverse correlation with MCV levels because these two sites have long been regarded as where active bone marrow resides. In contrast, total hip was mainly composed of cortical bone and few marrow compartments [23]. More than 60 years earlier, the distribution of actively proliferating bone marrow in adults was investigated through weighing and the visual inspection of bones from cadavers. These early data revealed most of the active red bone marrow is in the skull, ribs, vertebra, pelvis, and proximal extremities [24,25]. In 2011, Hayman et al. used 18F-fluoro-l-deoxythymidine positron emission/computed tomography to determine the location of proliferating bone marrow in 13 patients, and some of their results were similar to the previous data. Through a more modern estimation, they reported pelvis, thoracic spine, lumbar spine, sacrum, and proximal femurs had a higher mean percentage of proliferating bone marrow [26]. Campbell et al. observed similar findings using the same imaging modality to determine proliferating bone marrow percentages in non-small-cell lung cancer patients [27]. Finally, compared to femoral neck and lumbar spine, no significant association between total hip and MCV was detected. This may have resulted from total hip area being located at a more distal part of the femur, where red bone marrow would gradually metamorphose into inactive yellow marrow as the body ages [28].

Parathyroid hormone plays a vital role in vitamin D and phosphate metabolism through the modulation of the FGF23-Klotho axis [29]. Secondary hyperparathyroidism is commonly found in patients on dialysis, and the dysregulated hormone contributes to accelerated bone turnover and suppressed osteoblast formation, which often leads to osteoporosis development [30,31]. Apart from inducing bone fragility in CKD stage 5D patients, the negative association of the parathyroid hormone with hemoglobin has also been reported in CKD patients [32,33]. Hyperparathyroidism has been proposed to cause renal anemia through several mechanisms, including bone marrow fibrosis, interference with erythropoetin synthesis, and inhibition of bone marrow erythroid progenitors [34,35]. In view of its harmful effects on the differentiation of both osteoblasts and erythrocyte, hyperparathyroidism may be an alternative explanation for causing an inverse relationship between MCV and T-score observed in the present study.

Notably, among patients with CKD stage 5D, a significant inverse correlation between MCV and BMD was observed in patients on HD but not in patients on PD. Previous research has demonstrated that after 1 year of follow-up, patients on PD did not exhibit any decrease in total BMD or the BMD of any bone site, whereas patients on HD exhibited reduced BMD in almost every body part (i.e., total BMD, leg, trunks, rib, pelvis, and spine) The authors of that retrospective cohort study proposed that the continuous nature of PD, which does not lead to fluctuations in fluid status and circulating solutes, plays a role in the reduced risk of osteoporosis among patients on PD [36,37].

## 5. Limitations

This cross-sectional study had several limitations. First, only 123 participants were enrolled; thus, they may not represent all Taiwanese patients with CKD stage 5D. We encouraged each patient to take iron, calcium, and vitamin D supplements on a daily basis but failed to record the actual conditions of how participants took these supplements, which is another limitation of this study [30]. In addition, daily physical activity capacity and serum vitamin D concentration of each participant were not assessed, which are potential covariates that could substantially influence BMD. Furthermore, information was lacking regarding factors that could affect RBC size, including thyroid function, folic acid level, and vitamin B12 level. Furthermore, clinical variables that could influence BMD, including steroid usage, phosphate binders, bisphosphate, inflammation markers, bone turnover markers, and FGF23, were not measured in the present study; therefore, it was challenging to make full adjustments to the results for the potential confounding effects of the aforementioned variables. Finally, this study was limited by its cross-sectional design. More prospective studies with long-term follow-ups are required to determine whether MCV level is reliable for predicting the clinical prognosis of the CKD stage 5D population on chronic dialysis, particularly patients on HD.

## 6. Conclusions

In the CKD stage 5D population, BMD at bilateral femoral head and lumbar spine 1-4 were inversely correlated with MCV levels. Lumbar spine and femoral neck contained most of the active red bone marrow in the body, and BMD of these two sites was negatively correlated with MCV, suggesting a destructive stimulatory relationship between HSCs and osteoblasts. Significant inverse association between T-score and MCV levels was also observed after adjustments for different confounding factors. Further research on the underlying pathophysiological mechanism is required.

## Figures and Tables

**Figure 1 medicina-58-00497-f001:**
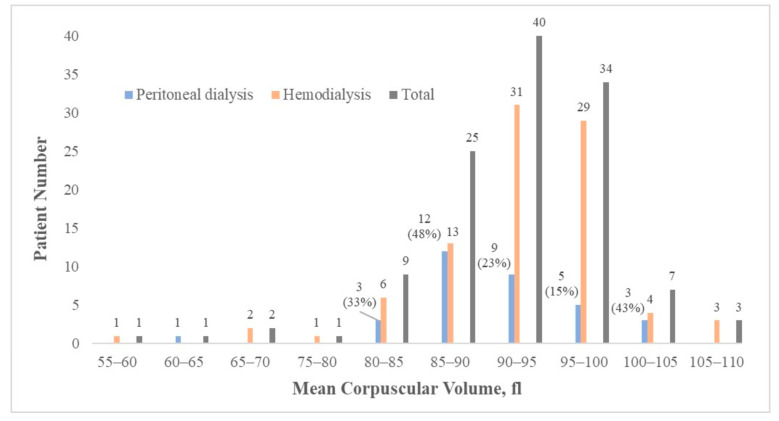
Patient distribution by MCV level and dialysis modalities.

**Table 1 medicina-58-00497-t001:** Baseline clinical characteristics of the study population and subgroup analysis between HD or PD.

Clinical Characteristics (*n* = 123)	Mean ± SD/Number (Percentage)	HD (*n* = 90)	PD (*n* = 33)	*p*-Value between HD and PD Groups
**Age**	65 ± 11	66 ± 10	62 ± 12	0.05
**Gender**				1.0
Male	61 (49.6%)	45 (50%)	16 (48%)	
Female	62 (50.4%)	45 (50%)	17 (52%)	
**BMI**	24 ± 4.1	24 ± 4.3	25 ± 3.6	0.5
**Mean duration of dialysis** (months)	59 ± 37	62 ± 38	50 ± 33	0.1
Types of dialysis		NA	NA	
Hemodialysis	90 (73%)			
Peritoneal dialysis	33 (27%)			
**Erythropoeisis-stimulating agent usage**	120 (98%)			
**Underlying comorbidities**				
Hypertensions	99 (80%)	75 (83%)	24 (73%)	0.2
Diabetes mellitus	63 (51%)	51 (57%)	12 (36%)	0.07
Dyslipidemia	41 (33%)	30 (33%)	11 (33%)	1.0
Gastrointestinal tract ulcer history	21 (17%)	17 (19%)	3 (9.1%)	0.3
**Smoking history**				0.4
Quitted	16 (13%)	8 (9%)	2 (6.1%)	
Current smoker	10 (8%)	9 (10%)	6 (18%)	
**Laboratory parameters**				
Hemoglobin, g/dL	10 ± 1.0	10 ± 1.0	10 ± 1.0	0.2
RBC ×10^6^/μL	3.4 ± 0.5	3.4 ± 0.5	3.3 ± 0.4	0.4
MCV, fl	92 ± 7.3	93 ± 7.4	91 ± 6.8	0.2
RDW, %	15 ± 2.3	15 ± 2.5	15 ± 1.5	0.6
WBC count ×10^3^/L	6.6 ± 2.0	6.4 ± 1.7	7.1 ± 2.6	0.11
Albumin, g/dL	3.8 ± 0.3	3.8 ± 0.3	3.6 ± 0.4	** <0.001 **
Phosphate, mg/dL	5.0 ± 1.6	5.1 ± 1.7	4.7 ± 1.3	0.3
Ca, mg/dL	9.1 ± 0.66	9.1 ± 0.7	9.3 ± 0.6	0.2
Corrected Ca, mg/dL	9.3 ± 0.74	9.2 ± 0.8	9.6 ± 0.6	** 0.02 **
Iron, ug/dL	66 ± 2	61 ± 25	80 ± 31	** 0.001 **
Ferritin, ng/mL	474 ± 369	455 ± 285	551 ± 544	0.2
Cholesterol, mg/dl	153 ± 37	152 ± 40	155 ± 27	0.7
GOT, U/L	17 ± 7.1	17 ± 7.4	17 ± 6	0.5
GPT, U/	14 ± 8.2	14 ± 7.8	14 ± 9.5	0.9
**Parathyroid hormone level** (pg/mL)	418 ± 494	381 ± 519	514 ± 414	0.2

Abbreviations: BMI—body-mass index; Ca—calcium; GOT—glutamate oxaloacetate transaminase; GPT—glutamate pyruvic transaminase; HD—hemodialysis; MCV—mean corpuscular volume; PD—peritoneal dialysis; RDW—red cell distribution width.

**Table 2 medicina-58-00497-t002:** Dual-energy X-ray absorptiometry results of the study population.

Clinical Characteristics	Mean ± SD/Number (Percentage)	HD (*n* = 90)	PD (*n* = 33)	*p*-Value between HD and PD Groups	*p*-Value of Correlation with MCV	Pearson Correlation Index
**T-score** (*n* = 123, the lowest T-score of the 5 sites)	−2.5 ± 1.2	−2.5 ± 1.3	−2.4 ± 0.9	0.8	**0.003**	** −0.27 **
**Femoral neck BMD**						
Right side (*n* = 120)	0.75 ± 0.16	0.75 ± 0.16	0.77 ± 0.14	0.5	**0.008**	** −0.24 **
Left side (*n* = 118)	0.74 ± 0.14	0.74 ± 0.15	0.74 ± 0.11	0.8	**0.02**	** −0.21 **
**Total hip BMD**						
Right side (*n* = 120)	0.82 ± 0.17	0.81 ± 0.18	0.84 ± 0.13	0.4	0.1	−0.15
Left side (*n* = 118)	0.82 ± 0.18	0.82 ± 0.17	0.81 ± 0.20	0.7	0.08	−0.16
**Lumbar spine 1–4 BMD** (*n* = 122)	1.14 ± 0.23	1.15 ± 0.25	1.11 ± 0.16	0.3	**0.038**	** −0.19 **
**Fracture Risk Assessment Tool**						
10-year probability of major osteoporotic fracture, %	11 ± 9.1	11.7 ± 10	9.0 ± 6.1	0.15	0.3	
10-year probability of hip fracture, %	5.4 ± 7.4	5.8 ± 8.1	4.2 ± 4.1	0.3	0.6	

Abbreviations: BMD—bone mineral density; HD—hemodialysis; MCV—mean corpuscular volume; PD—peritoneal dialysis; SD—standard deviation.

**Table 3 medicina-58-00497-t003:** Multivariate linear regression assessing the correlation between T-score and MCV among Taiwanese dialysis patients.

Total Patients (*n* = 123)	Variables	β	*p*-Value
Model 1 (MCV was set as the dependent variable)	T-score	−0.96	0.037
	Age	0.1	0.06
	Phosphate	−0.4	0.2
	RDW	−1.6	<0.001
	Hb	0.7	0.2
	WBC	−0.3	0.2
	Iron	0.007	0.7
	Ferritin	0.002	0.1
	Cholesterol	0.02	0.2
Model 2 (T-score was set as the dependent variable)	MCV	−0.027	0.048
	Age	−0.03	0.001
	Phosphate	0.07	0.3
	BMI	0.06	0.007
	Parathyroid hormone level	−0.001	0.01
	Dyslipidemia (Yes)	−0.3	0.5
	GI tract ulcer history (Yes)	0.5	0.3
Model 3 (T-score was set as the dependent variable)	MCV	−0.03	0.036
	Age	−0.03	0.002
	WBC	−0.05	0.4
	Albumin	0.2	0.6
	GOT	−0.02	0.3
	GPT	0.02	0.9

Model 1: MCV was set as the dependent variable and adjusted for T-score, age, serum phosphate level, RDW, Hb, WBC, iron, ferritin, and cholesterol. Model 2: T-score was set as the dependent variable and adjusted for MCV, age, serum phosphate level, BMI, parathyroid hormone level, dyslipidemia, and gastrointestinal tract ulcer history. Model 3: T-score was set as the dependent variable and adjusted for MCV, age, WBC, albumin, GOT, and GPT. Abbreviations: BMI—body-mass index; GOT—glutamate oxaloacetate transaminase; GPT—glutamate pyruvic transaminase; Hb—hemoglobin; MCV—mean corpuscular volume; RDW—red blood distribution width; WBC—white blood cell.

## Data Availability

Due to the sensitive nature of the questions asked in this study, survey respondents were assured that raw data would remain confidential and would not be shared.

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
