# Peer review of "Inverse Relationship between Mean Corpuscular Volume and T-Score in Chronic Dialysis Patients"

_medicina, 2022, doi:10.3390/medicina58040497_

Round 1

Reviewer 1 Report

The manuscript is interesting, however, there are issues that need to be addressed.

Major concerns

The study examined the relationship between T-scores and MCV in dialysis population. The authors chose to analyzed only the lowest T-Score of these 3 sites including total hip, femoral neck and lumbar spine. The detailed data on the BMD of each site and the relationship with MCV were not presented. Although both hip and lumbar spine are composed of cancellous bone, there exist differences in the changes of bone density in the two area.

Other issues

  1. The introduction is too long. since this is a clinical research, the introduction should focus more on clinical relationship between anemia and osteoporosis in ESKD. The first paragraph in the introduction is mostly irrelevant and should be deleted
  2. In the methods part, how was eGFR determined? Since these patients are on dialysis and serum creatinine fluctuates significantly before and after hemodialysis and. The levels are also different between HD and PD populations due to the effect of each dialysis modality on Cr clearance. I would suggesting removing all data on the eGFR. The data on residual kidney function analyzed from residual urine output is more relevant if such data is available
  3. Table 1 is too long. Please remove the data on eGFR as well as other non-relevant baseline data (gout, viral hepatitis, PKD etc.) The data on non-smoker can be removed (keep only the data current and past smoker). On laboratory data, consider removing serum Na, K, uric acid and triglyceride)
  4. Tables 2 should include the raw data on BMD for total hip, femoral neck and lumbar spine in addition to T-Score. The data on the relationship between BMD at each site and MCV are also required.

Please also state in the Table that the T-score used is the lowest T-score of the 3 sites.

  1. Figure 1 should contain percentage of the patients as well as the number to better elucidate the proportion of patients with each level of MCV. The 3rd bars represent all patients (HD +PD) should also be added.
  2. Supplementary Table 1 contains important data and should be included in the main manuscript. Again, please remove the data on eGFR as well as other non-relevant baseline data (gout, viral hepatitis, PKD etc.) The data on non-smoker can be removed (keep only the data current and past smoker). On laboratory data, consider removing serum Na, K, uric acid and triglyceride)
  3. In the text describing the findings in Table 3, the dependent variable for each model should be stated clearly. The way it is written now is confusing. The title of the table also requires corrections because the Table describes risk factors affecting both MCV and T-score and not T-score alone. The word “hemodialysis” in the title should be replaced with “dialysis” because this Table used data from all patients. Please remove eGFR from Model 2. Please clearly cite Table 3 in the texts.
  4. The Discussion is superfluous and way too long.

Please replace the word BMD with T-score since the study evaluated T-score and not BMD.

In the 3rd paragraph, in stead of discussion CKD-MBD, the authors should focus the discussion on the issue of BMD and anemia in dialysis population. Please also avoid using the word CKD-MBD but rather specify which component of CKD-MBD was being discussed (bone loss, electrolytes or hormone abnormalities???)

In the 4th paragraph, the relationship between FGF-23-klotho and bone loss is still far-fetched at this point. There are connections between FGF-23 and anemia which does not involve bone loss.  The authors should also discuss the issue related to PTH, bone marrow and anemia.

I could not find any discussion on the findings in Table 2. Please include the discussion relevant to the findings in Table 2.

The discussion on the differences between HD and PD should be brief since there are only 33 patients in the PD group and the data are presented in the supplementary tables and not the main findings of the study

  1. The conclusion should state only the main findings for the whole ESKD population. The speculation on the unfound mechanisms should be removed or revised. The difference among HD and PD should also be removed.
  2. Similarly for the abstract, the sentence “a subgroup analysis between HD and PD……………………” and “HD subgroup showed lower BMD was highly associated………………………………” as well as the speculation on the mechanisms in the conclusion should also be removed or revised to a more relevant aspect.

Author Response

Dear Reviewer 1,

Thank you for giving us the opportunity to revise our submitted manuscript. Thanks to the constructive review comments so that my colleagues and I are able to revise the manuscript substantially and resubmit it. We tried our best to improve the scientific contents. Our changes are in red font in the revised manuscript and clean edition was also attached as the author_coverletter-18063805.v1.doc.  Please inform us if any additional revisions are needed.

Major concerns

The study examined the relationship between T-scores and MCV in dialysis population. The authors chose to analyzed only the lowest T-Score of these 3 sites including total hip, femoral neck and lumbar spine. The detailed data on the BMD of each site and the relationship with MCV were not presented. Although both hip and lumbar spine are composed of cancellous bone, there exist differences in the changes of bone density in the two area.

Reply

Thank you for providing us with this precious advice. We have reassessed the data and added BMD of bilateral femoral heads, total hips, and lumbar spine 1-4 into Table 2. Correlation between MCV and BMD data were also conducted by Pearson correlation tests, which revealed MCV was significantly associated with BMD at bilateral femoral neck and lumbar spine 1-4. Further discussion was added in the Discussion section of the manuscript. (clean edition line: 289 - 307)

P.S. Following suggestions from the second reviewer, we have changed the term “ESKD” into “CKD stage 5D” throughout the whole manuscript.

Question 1:

The introduction is too long. since this is a clinical research, the introduction should focus more on clinical relationship between anemia and osteoporosis in ESKD. The first paragraph in the introduction is mostly irrelevant and should be deleted.

Reply 1:

Thank you for this precious suggestion. We removed the first paragraph which was irrelavant and shortened the Introduction section. The second paragraph was also revvised to address more on the potential mechanism of the correlation between anemia and osteoporosis in CKD stage 5D.

Question 2:

In the methods part, how was eGFR determined? Since these patients are on dialysis and serum creatinine fluctuates significantly before and after hemodialysis and. The levels are also different between HD and PD populations due to the effect of each dialysis modality on Cr clearance. I would suggesting removing all data on the eGFR. The data on residual kidney function analyzed from residual urine output is more relevant if such data is available

Reply 2:

All data associated with residual kidney function were removed from the manuscript and tables.  Analysis were reconducted.

Question 3:

Table 1 is too long. Please remove the data on eGFR as well as other non-relevant baseline data (gout, viral hepatitis, PKD etc.) The data on non-smoker can be removed (keep only the data current and past smoker). On laboratory data, consider removing serum Na, K, uric acid and triglyceride)

Reply 3:

Non-relevant data (including eGFR, gout, viral hepatitis, PKD, none smoking history, Na, K, uric acid, triglyceride, total iron binding capacity, platelet counts) in Table 1. were all removed.

Question 4:

Tables 2 should include the raw data on BMD for total hip, femoral neck and lumbar spine in addition to T-Score. The data on the relationship between BMD at each site and MCV are also required.

Please also state in the Table that the T-score used is the lowest T-score of the 3 sites.

Reply 4:

Data on BMD for bilateral femoral neck, total hip, and lumbar spine 1-4 were added to Table 2. Results and discussions about the correlation between BMD at these 5 sites and MCV were presented in the manuscript. (clean edition line: 194 – 198, 289 - 307)

Question 5:

Figure 1 should contain percentage of the patients as well as the number to better elucidate the proportion of patients with each level of MCV. The 3rd bars represent all patients (HD +PD) should also be added.

Reply 5:

We have reproduced a new Figure 1. Patient number and their respective percentage in each level of MCV as well as a  3rd bar were added.

Question 6:

Supplementary Table 1 contains important data and should be included in the main manuscript. Again, please remove the data on eGFR as well as other non-relevant baseline data (gout, viral hepatitis, PKD etc.) The data on non-smoker can be removed (keep only the data current and past smoker). On laboratory data, consider removing serum Na, K, uric acid and triglyceride)

Reply 6:

We have included the findings in supplementary table 1 by briefly summarizing them in the Result section of the manuscript (clean edition line: 211 - 215). In addition, we have removed all non-relevant data and eGFR value from the supplementary material.

Question 7:

In the text describing the findings in Table 3, the dependent variable for each model should be stated clearly. The way it is written now is confusing. The title of the table also requires corrections because the Table describes risk factors affecting both MCV and T-score and not T-score alone. The word “hemodialysis” in the title should be replaced with “dialysis” because this Table used data from all patients. Please remove eGFR from Model 2. Please clearly cite Table 3 in the texts.

Reply 7:

We are sorry for the confusing writing and inappropriate Table title. Revisions have been made regarding the text description on findings in Table 3 (clean edition line: 215 - 233). Moreover, title of Table 3 was changed to “Table 3. Multivariate linear regression assessing the correlation between T-score and MCV among Taiwanese dialysis patients.”.

Question 8:

The Discussion is superfluous and way too long.

Please replace the word BMD with T-score since the study evaluated T-score and not BMD.

In the 3rd paragraph, instead of discussion CKD-MBD, the authors should focus the discussion on the issue of BMD and anemia in dialysis population. Please also avoid using the word CKD-MBD but rather specify which component of CKD-MBD was being discussed (bone loss, electrolytes or hormone abnormalities???)

In the 4th paragraph, the relationship between FGF-23-klotho and bone loss is still far-fetched at this point. There are connections between FGF-23 and anemia which does not involve bone loss.  The authors should also discuss the issue related to PTH, bone marrow and anemia.

I could not find any discussion on the findings in Table 2. Please include the discussion relevant to the findings in Table 2.

The discussion on the differences between HD and PD should be brief since there are only 33 patients in the PD group and the data are presented in the supplementary tables and not the main findings of the study

Reply 8:

Thank you for these great suggestions on how to improve the contents of the Discussion section. We have critically revised it. First, the word “BMD” was replaced with “T-score” if the result was associated with T-score instead of BMD. Regarding that CKD-MBD was clearly not so relevant to the present study, all authors agreed to delete the whole 3rd paragraph. Further discussion on the findings in Table 2 was also added at paragraph 4. In terms of discussing the interaction of parathyroid hormone abnormalities and bone marrow and osteoporosis, a new paragraph was added (clean edition line: 308 - 320). At last, we have shortened the discussion on the difference between HD and PD since it was not a main finding of this study.

Question 9:

The conclusion should state only the main findings for the whole ESKD population. The speculation on the unfound mechanisms should be removed or revised. The difference among HD and PD should also be removed.

Reply 9:

The conclusion section was rewritten and only included the main findings of the present study. Sentences related with HD and PD subgroup analysis were removed and speculation on the unproven mechanisms were revised.

Question 10:

Similarly for the abstract, the sentence “a subgroup analysis between HD and PD……………………” and “HD subgroup showed lower BMD was highly associated………………………………” as well as the speculation on the mechanisms in the conclusion should also be removed or revised to a more relevant aspect.

Reply 10:

The abstract section was revised, and all sentences related with subgroup analysis were removed. (clean edition line: 59 - 77)

Reviewer 2 Report

Interesting work. Few suggestions:

It would be desirable to use the term CKD and CKD stage 5 D instead of ESKD. 

The diagnosis of osteoporosis in CKD is not simple ( open question if the WHO criteria is the same for dialysis population as for general population ) there it may be better to use the term CKD MBD with a decrease of bone mass , i.e. T score.

It would be useful to know the treatment of pts with vitamin D or analogues, calcimimetics as well with iron supplements. 

Author Response

Dear Reviewer 2,

Thank you for giving us the opportunity to revise our submitted manuscript. Thanks to the constructive review comments so that my colleagues and I are able to revise the manuscript and resubmit it. We tried our best to improve the scientific contents. Our changes are in red font in the revised manuscript and clean edition was also attached as the author_coverletter-18194585.v1.doc.  Please inform us if any additional revisions are needed.

Question 1:

Interesting work. Few suggestions:

It would be desirable to use the term CKD and CKD stage 5 D instead of ESKD.

Reply 1:

  Thank you so much for your comments and suggestions. We have changed the term “ESKD” into “CKD stage 5D” throughout the whole manuscript.

Question 2:

The diagnosis of osteoporosis in CKD is not simple ( open question if the WHO criteria is the same for dialysis population as for general population ) there it may be better to use the term CKD MBD with a decrease of bone mass , i.e. T score.

Reply 2:

We have checked the WHO osteoporosis diagnosis criteria published in 1994 and the National Institute of Health, but both guidelines did not mention any specific osteoporosis definitions for dialysis population 1. Making diagnosis of osteoporosis in CKD patients is indeed difficult. In the chapter “Osteoporosis in patients with chronic kidney disease: Diagnosis and evaluation” from UpToDate, it sated that in CKD 5 population, a diagnosis of osteoporosis can only be made by excluding CKD-MBD or renal osteodystrophy. However, in the latest “European Consensus Statement on the diagnosis and management of osteoporosis in chronic kidney disease stages G4-G5D”, they preferred to adopt the conventional osteoporosis definition for CKD patients because these patients were simultaneously influenced by primary and secondary osteoporosis and it is hard to determine whether renal osteodystrophy had more influence on bone health over primary causes 2. In clinical practice, it is reasonable to arrange DXA for BMD measurement for advanced CKD patients because numerous studies have proven that low BMD is indeed strongly associated with fracture risk. Regarding CKD MBD are characterized by low bone turnover rate and abnormal mineralization, it is also important to routinely follow up bone turnover biomarkers in this specific population (ex: parathyroid hormone, alkaline phosphatase, bone-specific alkaline phosphatase) 3.

Reference 1, NIH Consensus Development Panel on Osteoporosis Prevention, Diagnosis, and Therapy. Osteoporosis prevention, diagnosis, and therapy. JAMA 2001; 285: 785–795

Reference 2, Evenepoel P, Cunningham J, Ferrari S, Haarhaus M, Javaid MK, Lafage-Proust MH, Prieto-Alhambra D, Torres PU, Cannata-Andia J; European Renal Osteodystrophy (EUROD) workgroup, an initiative of the CKD-MBD working group of the ERA-EDTA, and the committee of Scientific Advisors and National Societies of the IOF. European Consensus Statement on the diagnosis and management of osteoporosis in chronic kidney disease stages G4-G5D. Nephrol Dial Transplant. 2021 Jan 1;36(1):42-59.

Reference 3, Charles Ginsberg, Joachim H. Ix, Diagnosis and Management of Osteoporosis in Advanced Kidney Disease: A Review, American Journal of Kidney Diseases, Volume 79, Issue 3, 2022; 427-436

Question 3:

It would be useful to know the treatment of pts with vitamin D or analogues, calcimimetics as well with iron supplements.

Reply 3:

We are sorry that we did not ask and record patient clinical history of taking these supplementations even though it is a routine practice for us to encourage every patient to take these supplementations. Therefore, we have listed it as a limitation of this study. (clear edition line: 333 - 335)

Round 2

Reviewer 1 Report

The authors have satisfatorily respnded to all questions and concerns.